# Metabolomics Reveals Changes in Metabolite Profiles among Pre-Diapause, Diapause and Post-Diapause Larvae of *Sitodiplosis mosellana* (Diptera: Cecidomyiidae)

**DOI:** 10.3390/insects13040339

**Published:** 2022-03-30

**Authors:** Qitong Huang, Qian Ma, Fangxiang Li, Keyan Zhu-Salzman, Weining Cheng

**Affiliations:** 1Key Laboratory of Plant Protection Resources & Pest Management of the Ministry of Education, College of Plant Protection, Northwest A&F University, Yangling 712100, China; 13760637249hqt@nwafu.edu.cn (Q.H.); maqian98@nwafu.edu.cn (Q.M.); 2Xi’an Agricultural Technology Extension Centre, Xi’an 710061, China; lfx070707@163.com; 3Department of Entomology, Texas A&M University, College Station, TX 77843, USA

**Keywords:** *Sitodiplosis mosellana*, pre-diapause, diapause, post-diapause, metabolomics, stress tolerance

## Abstract

**Simple Summary:**

Diapause is a programmed developmental arrest coupled with an evident reduction in metabolic rate and a dramatic increase in stress tolerance. *Sitodiplosis mosellana*, a periodic but devastating wheat pest, spends the hot summer and cold winter as diapausing larvae. However, little is known about the metabolic changes underlying this obligatory diapause. The objective of this study was to identify significantly altered metabolites and pathways in diapausing *S. mosellana* at stages of pre-diapause, diapause, post-diapause quiescence and post-diapause development using gas chromatography/time-of-flight mass spectrometry and the orthogonal partial least squares discriminant analysis. Pairwise comparisons of the four groups showed that 54 metabolites significantly changed. Of which, 37 decreased in response to diapause, including four TCA cycle intermediates and most amino acids, whereas 12 increased. Three metabolites were significantly higher in the cold quiescence stage than in other stages. The elevated metabolites included the well-known cryoprotectants trehalose, glycerol, proline and alanine. In conclusion, the low metabolic rate and cold tolerance *S. mosellana* displayed during diapause may be closely correlated with its reduced TCA cycle activity or/and the increased biosynthesis of cryoprotectants. The results have contributed to our understanding of the biochemical mechanism underlying diapause and the related stress tolerance in this key pest.

**Abstract:**

*Sitodiplosis mosellana*, a notorious pest of wheat worldwide, copes with temperature extremes during harsh summers and winters by entering obligatory diapause as larvae. However, the metabolic adaptive mechanism underlying this process is largely unknown. In this study, we performed a comparative metabolomics analysis on *S. mosellana* larvae at four programmed developmental stages, i.e., pre-diapause, diapause, low temperature quiescence and post-diapause development. In total, we identified 54 differential metabolites based on pairwise comparisons of the four groups. Of these metabolites, 37 decreased in response to diapause, including 4 TCA cycle intermediates (malic acid, citric acid, fumaric acid, α-ketoglutaric acid), 2 saturated fatty acids (palmitic acid, stearic acid) and most amino acids. In contrast, nine metabolites, including trehalose, glycerol, mannitol, proline, alanine, oleic acid and linoleic acid were significantly higher in both the diapause and quiescent stages than the other two stages. In addition to two of them (trehalose, proline), glutamine was also significantly highest in the cold quiescence stage. These elevated metabolites could function as cryoprotectants and/or energy reserves. These findings suggest that the reduced TCA cycle activity and elevated biosynthesis of functional metabolites are most likely responsible for maintaining low metabolic activity and cold tolerance during diapause, which is crucial for the survival and post-diapause development of this pest.

## 1. Introduction

Diapause is a genetically programmed state of developmental arrest that can occur at the embryonic [1], larval [2], pupal [3] or adult stage of a specific insect species [4]. A hallmark of diapause is metabolic suppression, which enables insects to survive adverse seasons and synchronize their life cycles with periods that are suitable for growth, development and reproduction [5,6]. Moreover, diapause is usually associated with an enhanced stress tolerance through the biosynthesis of some protective molecules (e.g., glycerol, sorbtiol, trehalose) [7,8]. Synthesis of these compounds is promoted by diapause initiation [9,10] and/or is triggered by low temperatures during diapause [11,12]. Due to its central role in insect survival and reproduction, a better understanding of the biochemical adaptation mechanism underlying diapause is crucial for the development of management strategies for economically significant pests [13].

The orange wheat blossom midge *Sitodiplosis mosellana* Géhin (Diptera: Cecidomyiidae) is a periodic but devastating pest of wheat *Triticum aestivum* L. (Poaceae). It feeds on developing wheat kernels, causing tremendous losses in yield and economics [14,15,16]. This univoltine midge undergoes an obligatory larval diapause in soil inside cocoons. Mature third-instar larvae drop from wheat ears to the ground in mid-to-late May in most northern China regions and burrow into the soil to spin cocoons to initiate diapause. They stay in cocoons until March of the next year [17], but actually terminate diapause by chilling stimulation in December and enter the low-temperature quiescent stage thereafter [18]. In the spring, they leave the cocoons to start post-diapause development in response to the rising ambient temperature. In other words, the long period of diapause and quiescence enables this species to cope with harsh summer and winter conditions and aligns its development with the wheat phenology. We thus propose that the concentrations of some metabolites could drop in both the diapause and quiescent stages due to reduced metabolic activity, whereas some other metabolites might be accumulated for energy reserves or to increase stress tolerance. Previous studies on the biochemical basis of diapause in *S. mosellana* have mainly focused on a few routine compounds (i.e., glycerol, trehalose, triglyceride and fatty acids) [19,20,21,22] and the global metabolic change remains to be investigated.

Metabolomics, a rapidly developing technology of the post-genomics era, involves a holistic analysis of all low molecular weight metabolites in an organism or specific cells. The ability to profile metabolites during various growth conditions or under certain stresses [23,24,25] makes it possible to identify putative biochemical pathways related to various physiological processes. Currently, gas chromatography/time-of-flight mass spectrometry (GC/TOF-MS) [26], liquid chromatography/time- of-flight mass spectrometry (LC/TOF-MS) [27] and nuclear magnetic resonance (NMR) [28] are commonly used for metabolite detection and quantification. Among them, GC/TOF-MS has been used most extensively, owing to its high resolution, high detection sensitivity and numerous open-access spectral libraries [29]. It has been successfully employed in multiple aspects of entomology research, including diapause in *Sarcophaga crassipalpis* [30], *Corythucha ciliata* [31] and *Pieris napi* [32]. As a result, a number of candidate metabolites (e.g., inositol, mannitol, alanine, etc.) playing important roles during diapause have been identified.

To identify significantly altered metabolites and pathways during diapause of *S. mosellana*, in the present study, we applied GC/TOF-MS to profile metabolites of *S. mosellana* larvae at four programmed developmental stages, i.e., the pre-diapause, diapause, low temperature quiescence and post-diapause development stages. Further, we used multivariate statistical analysis to screen for differential metabolites and a KEGG enrichment analysis was performed. We believe that the results will facilitate our understanding of the biochemical adaptive mechanism underlying diapause and the related stress tolerance of this key wheat pest.

## 2. Materials and Methods

### 2.1. Insect Collection

*S. mosellana* third instar larvae at any one of the four physiological stages associated with diapause, including the pre-diapause, diapause, post-diapause quiescence and post-diapause development were collected according to the method described by Cheng et al. (2016) [33]. Briefly, pre-diapause larvae were collected by dissecting wheat spikes severely infested by *S. mosellana* in mid-May 2018. A large number of wheat ears containing pre-diapause larvae were synchronously gathered and placed on wet soil in a field insectary built in Northwest A&F University (Yangling, China) to enable insects to burrow into the soil. Diapause initiation is indicated by larvae forming cocoons. We found that almost all cocooned larvae collected in December or later could arouse and emerge as adults once exposed to a favorable temperature (25 °C), indicating that by December they had completed diapause and entered post-diapause quiescence. Cocooned diapausing and post-diapause quiescent larvae were collected from the insectary in late July and late December 2018, respectively. Post-diapause developing larvae (i.e., larvae exiting cocoons) were collected in late March 2019. For each developmental stage, about 150 larvae (40 mg) were grouped together as a sample and put into a 2.0 mL tube and at least six replicates of samples were prepared. All samples collected were first frozen in liquid nitrogen and then stored at −80 °C until metabolic analysis.

### 2.2. Metabolite Extraction and Derivatization

For each sample (about 150 larvae, 40 mg), metabolites were extracted and derivatized following a procedure described by Cui et al. (2017) [34] with slight modifications. Specifically, 600 μL of ice-cold extraction solution containing methanol: chloroform: ultrapure water (3:1:2 *v*/*v*/*v*), 10 μL of 0.3 mg/mL L-2-chlorophenylalanine (internal standard) and four steel beads (3 mm in diameter) were added to each sample. After 30 s of vortexing, the mixture was homogenized for 5 min using a TissueLyser (Qiagen, Hilden, Germany), vortexed for 1 min and sonicated for 20 min. The resulting extract was centrifuged for 10 min at 12,000× *g* rpm at 4 °C. Supernatant (400 μL) was transferred to a new GC/MS glass vial, concentrated for 2–3 h in a vacuum concentrator and dried completely using nitrogen. Dried samples were immediately methoxymated by adding 80 μL methoxylamine hydrochloride (15 mg/mL inpyridine) and incubated for 90 min at 37 °C, and then trimethylsilylated by adding 80 μL N,O-bis (trimethylsilyl) trifluoroacetamide (BSTFA) (containing 1% TMCS) and incubated for 1 h at 70 °C. The samples were kept at an ambient temperature (22–25 °C) for 30 min before GC/TOF-MS analysis.

### 2.3. GC/TOF-MS Analysis of Metabolites

One μL of each derivatized sample was injected in the splitless mode into an Agilent 7890A gas chromatography (GC) system coupled with a Pegasus 4D time-of-fight mass spectrometer (TOF/MS). The GC was equipped with a DB-5MS column (0.25 μm, 0.25 mm × 30 mm; J&W Scientific, Folsom, CA, USA). The injection temperature was 280 °C. Helium was used as a carrier gas with a flow rate of 1 mL/min. The column temperature was programmed with an initial temperature of 90 °C for 0.2 min, then increased to 240 °C at 5 °C/min, further increased to 300 °C at 25 °C/min and maintained at 300 °C for 14.4 min. The MS was used in the electron impact mode, with the ion source voltage of 70 ev. The transfer line and ion source temperatures were 280 °C and 220 °C, respectively. Data were collected from the full scan mode across a mass range of 33 to 550 *m*/*z* with a rate of 10 spectrum/s after a solvent delay of 420 s.

### 2.4. Data Preprocessing

Raw data obtained by GC/TOF-MS were first analyzed using ChromaTOF software (Version 4.34, LECO, St Joseph, MI, USA) for the baseline filtering, peak extraction, deconvolution analysis, noise reduction, peak alignment and peak area integration [35]. The noise was filtered by the interquartile range and only groups missing values no larger than 50% of peak area data were retained and filled up by a simulation method with half of the minimum value. The peaks were identified qualitatively by matching their mass spectra and retention time index with those of the LECO-Fiehn Rtx5 database. Peak areas were normalized to the integral standard (L-2-chlorophenylalanine) signal.

### 2.5. Multivariate Analysis

Normalized GC/TOF-MS data were imported into SIMCA software package (version 14.0, Umetrics, Umeå, Sweden) for multivariate statistical analyses. The principal component analysis (PCA) was firstly performed to visualize the distribution of the original data and general separation among experimental groups (pre-diapause, diapause, post-diapause quiescence and post-diapause development). The orthogonal partial least squares discriminant analysis (OPLS-DA) was subsequently applied to obtain maximal group separation and cross-validated by 200 permutation tests. The variable importance in the projection (VIP) value from the OPLS–DA model (VIP > 1) combined with Student’s *t*-test (*p* < 0.05) and the fold change (FC > 2 or <0.5) were used to determine significantly different metabolites between pairwise comparisons of the four groups. The FC of each metabolite was calculated by comparing the mean values of the normalized peak intensities obtained. 

Based on differential metabolite data, a two-way heatmap with hierarchical clustering was constructed using the online platform (http://www.bioinformatics.com.cn) (accessed on 10 December 2021). Kyoto Encyclopedia of Genes and Genomes (KEGG) pathway enrichment analysis was conducted using the OmicStudio tools (https://www.omicstudio.cn/tool) and *p* < 0.05 was considered significant. Differences in the relative quantities of differential metabolites among the four groups were also analyzed by one-way analysis of variance followed by Tukey’s multiple range test (*p* < 0.05). Analyses were carried out with SPSS version 20.0 (Chicago, IL, USA).

## 3. Results

### 3.1. General Metabolite Profiles of S. mosellana Larvae

A total of 94 metabolites were identified using GC-TOF/MS across all samples (6 × 4) derived from *S. mosellana* third instar larvae at four physiological stages associated with diapause including pre-diapause (PreD), diapause (D), post-diapause quiescent (PDQ) and post-diapause development (PDD). These identified metabolites belong to six distinct biochemical types: amino acids (29), sugars and polyols (22), tricarboxylic acid cycle (TCA) intermediates (4), free fatty acids (4), organic acids (16) and others (19) (Appendix A).

### 3.2. PCA and OPLS-DA Analyses of Metabolic Difference

Principal component analysis (PCA) of the 94 metabolites showed clear separations between PreD/PDD and other two stages (D, PDQ), as well as between PreD and PDD in the score plots among four groups (Figure 1) and in pairwise comparisons (Appendix A). In contrast, D and PDQ tended to group separately in the score plot of pairwise comparison although it was less obvious in the four-group plot. All samples in the score plots were located in the 95% Hotelling’s T2 ellipse, indicating that no outliers were present in the analyzed samples and all samples could be used in the following analysis.

To further display the difference in the metabolite profiles between groups, pairwise comparative OPLS-DA was conducted with one orthogonal (PC2) and one predictive component (PC1) for all models derived from two groups of samples. As shown in Figure 2A–F, a clear separation in the score plot was obtained between all pairwise comparisons (Figure 2D). Moreover, model parameters for the interpretability (R^2^) and the predictability (Q^2^) from the software were all higher than 0.93, indicating excellent model quality. Moreover, the R^2^ intercept values on the Y axis determined after 200 permutation tests (Figure 2A–F) were lower than the R^2^ values in the corresponding models, and the Q^2^ intercept on the Y axis was less than zero, suggesting a low risk of over-fitting and reliability of the models. Taken together, the OPLS-DA models were capable of separating all pairwise groups.

### 3.3. Identification of Differential Metabolites among Developmental Stages

Based on the VIP scores from the OPLS–DA models (VIP > 1), student’s *t*-test (*p* < 0.05) and fold change (FC > 2.0 or <0.5), 54 out of 94 metabolites, that contributed the most to the variation of metabolic profiles were identified as differential metabolites in six pairwise comparisons (Figure 3). They included 20 amino acids, 19 sugars and polyols, 4 TCA intermediates, 4 fatty acids, 5 organic acids and 2 others. Among them, only 10 varied between D and PDQ and 9 of them were higher in PDQ. When these two stages were compared to PreD, 37 and 35 metabolites decreased and 12 and 14 metabolites increased in D and PDQ, respectively. Relative to the PDD, 38 and 34 metabolites were lower and 10 and 11 metabolites were higher in D and PDQ, respectively. The levels of 34 metabolites were different between PreD and PDD. Details of these differential metabolites can be seen in the two-way heat-map with a hierarchical clustering diagram (Figure 4) and Appendix A.

### 3.4. Alteration of Metabolic Pathways during Diapause

To determine the potential metabolic pathways altered during diapause, KEGG pathway annotation and enrichment analysis for three pairwise comparisons, including “D vs. PreD” (Figure 5A), “PDQ vs. D” (Figure 5B) and “PDD vs. PDQ” (Figure 5C), were performed based on the identified DMs. Figure 5 shows the top 20 pathways with the most significant impact for each pairwise comparison. Compared to D, the major metabolic pathways that differed in PDQ were those underlying the metabolisms of amino acids (“alanine, aspartate and glutamate metabolism”, “arginine and proline metabolism”, “glutamine and glutamate metabolism”) and carbohydrates (i.e., “TCA cycle”, “pentose phosphate pathway”). Besides these metabolic changes, pathways of “glycine, serine and threonine metabolism”, “glycolysis”, “starch and sucrose metabolism” and “biosynthesis of unsaturated fatty acids” were also altered dramatically in the other two comparisons.

### 3.5. Change in Key Metabolite Levels during Diapause

To better explore the roles of differential metabolites identified during *S. mosellana* diapause, their relative quantity in four larval stages related to diapause (i.e., PreD, D, PDQ and PDD) were analyzed together. Figure 6 showed changes in the levels of representative metabolites. Compared to both PreD and PDD, 27 metabolites were significantly lower at both D and PDQ. These decreased metabolites included nine amino acids (e.g., serine, glycine, allothreonine, threonine), nine carbohydrates (e.g., glucose, fructose 1,6-bisphosphate, xylitol), all four TCA cycle intermediates (malic acid, citric acid, fumaric acid, α-ketoglutaric acid), two saturated fatty acids (palmitic acid, stearic acid), two organic acids (lactic acid, 6-phosphoglyconic acid) and urea (Figure 4). In marked contrast, nine metabolites, including two amino acids (i.e., alanine, proline), five carbohydrates (i.e., trehalose-6-phosphate, trehalose, glycerol, mannitol, palatinitol) and two unsaturated fatty acids (i.e., oleic acid, linoleic acid), had significantly higher levels at both D and PDQ. Clearly, the levels of trehalose, proline and glutamine were significantly higher in cold PDQ than the other three stages. Furthermore, mannitol and trehalose were most abundant in both D and PDQ, especially mannitol in D and trehalose in PDQ. Pyruvic acid and ribulose 5-phosphate levels greatly increased in D, continuously rose and reached a peak in PDD (Figure 4). Based on these key differential metabolites, we summarized the related metabolic network changes during *S. mosellana* diapause (Figure 7).

## 4. Discussion

Differing from many biochemical studies on insect diapause, the insect samples used in this study (i.e., pre-diapause, diapause, post-diapause quiescence and post-diapause developmental *S. mosellana* larvae) were collected from the field. The results should thus be more relevant to the natural situation. As expected, a comparative metabolomic analysis among the four stages revealed some significantly altered metabolites/pathways during diapause, which potentially affect the survival and reproduction of this pest. These metabolites/pathways primarily involve low molecular weight sugars and polyols (e.g., trehalose, glycerol, mannitol), TCA cycle intermediates, amino acid metabolism, pentose phosphate pathway (PPP) and the biosynthesis of unsaturated fatty acids. The potential functions of some metabolites/pathways during the diapause of this midge and other insect pests are discussed as follows.

Elevated levels of low molecular weight sugars and polyols are commonly observed in diapausing arthropods as a means to endure adverse environments and likewise, glycerol and trehalose are the best-known molecules that increase during diapause or cold acclimation [12,36]. They may serve as important energy reserves for post-diapause development, as well as cryoprotectants by stabilizing proteins and membranes and reducing the body’s supercooling point [31,37]. Consistently, a massive accumulation of glycerol and trehalose occurred in diapause and particularly in the cold quiescence stage of *S. mosellana*. The great increase in trehalose-6-phosphate (Figure 4), the precursor of trehalose synthesis [38] and the dramatic reduction in the digestive product of trehalase (i.e., glucose) after diapause may be conducive to trehalose accumulation. The observed increase in glycerol may originate from the transformation of triglyceride, which significantly decreased after diapause [22]. Notably, palatinitol was found to exhibit a similar change pattern with that of glycerol and trehalose in this study (Figure 4), implying their similar function during *S. mosellana* diapause. To our knowledge, this is the first report that this polyol may be related to insect diapause.

Mannitol has been reported as not only a cryoprotectant along with other sugars and polyols during over-wintering or diapausing periods in some insects [39,40], but also as a thermo-osmoprotectant in response to the high temperature/drought stress [41,42]. Our study here revealed that, among all the metabolites identified, mannitol is the most abundant in over-summering diapause larvae and the second most abundant (next to trehalose) in over-wintering quiescence individuals, implying its important roles as a thermo-osmoprotectant in the heat and a cryoprotectant in the cold tolerance of *S. mosellana*, respectively. Xylitol, another reported polyol cryoprotectant [43], however, was extremely lower in both the diapause and cold quiescence stages (Figure 4). Hence, it is unlikely that xylitol plays any protective role in *S. mosellana*.

Metabolic rate depression is a typical feature of insect diapause [44] and this was also demonstrated in *S. mosellana* [45]. *S. mosellana* undergoes diapause in soil and we thus anticipated seeing signs of reduced mitochondrial energy production coinciding with diapause. As expected, the level of all four aerobic metabolic intermediates detected (i.e., malic acid, citric acid, fumaric acid, α-ketoglutaric acid) significantly decreased during the diapause and quiescence stages, indicating a reduction in TCA cycle activity. Similar results have also been reported in several recent studies for other arthropods species including the flesh flies *S. crassipalpis* [30], the aphid parasitoid *Praon volucre* [43] and the two-spotted spider miter *Tetranychus urticae* [46]. These results suggest that the depression of TCA cycle activity may be responsible for metabolic suppression during diapause at least for these species, which presumably conserves nutrient reserves. We also observed the low levels of glycolysis intermediate fructose 1,6-bisphosphate in diapause. Such a result is in accordance with the reduced activity of phosphofructokinase (PFK), an essential enzyme in glycolysis, in diapausing *S. mosellana* [47] and *Drosophila suzukii* [48]. The level of pyruvic acid, the end-product of glycolysis and the starting substrate for TCA cycle, however, increased after diapause. This is most likely the result of a rapid reduction in the TCA cycle activity and some amino acid metabolism as discussed below. Increased pyruvate levels during diapause have several potential effects: (1) acting as a readily available energy source for the resumed TCA cycle activity at the end of diapause [49]; (2) serving as a building block for the synthesis of amino acids, especially alanine.

The pentose phosphate pathway (PPP), an alternative route of glucose catabolism, is a major source of the reducing power (i.e., NADPH) for biosynthetic processes [50,51]. It has been reported that PPP activity increases with decreasing temperature [52] and also in diapausing *T. urticae* [46]. The reduction in 6-phosphogluconic acid and the significant increase in ribulose 5-phosphate after diapause (Figure 4) indicated the activation of PPP, resulting in large amounts of NADPH which can then be used for biosynthesis of unsaturated fatty acids (UFAs). To maintain the fluidity and integrity of the membrane, diapausing or cold-acclimated insects generally possess elevated quantities of UFAs [53,54,55]. This may be supported by greater levels of oleic acid (C18:1) and linoleic acid (C18:2) and lower levels of palmitic acid (C16:0) and stearic acid (C18:0) in both the diapause and quiescence stages (Figure 4 and Figure 6).

Amino acid metabolism is altered dramatically during diapause. Most amino acids decrease in response to diapause and some of these amino acids, including serine, glycine, allothreonine and threonine, can be used to synthesize pyruvate [25,56], implying that their decreased levels are likely correlated with an increased pyruvate level after diapause. This result agrees with previous studies showing that serine and glycine levels are lower in diapausing *T. urticae* adults [46] and diapausing *Ostrinia nubilalis* larvae [57]. Well-known insect cryoprotectants alanine and proline were previously reported to be increased in response to diapause and low temperatures in several other species [58,59,60], consistent with our findings in this study. Elevated alanine has been demonstrated to exert synergistic and colligative effects with other solutes on cold tolerance [61] but may also serve as an alternative and less toxic by-product of pyruvate catabolism than lactic acid [62], as evidenced by the decrease in lactic acid during diapause (Figure 4). Notably, glutamine accumulated considerably in the cold quiescence stage, indicating its importance in cold tolerance. Similar results have been reported for the flesh fly *Sarcophaga crassipalpis* [30] and in the diapausing cricket *Teleogryllus commodus* [63]. In addition, glutamine has the non-colligative potential for enhancing the responsiveness of heat shock proteins (Hsps) [64]. Thus, the up-regulated expression of Hsp17.4, Hsp20.3 and Hsp90 genes in *S. mosellana* in winter [33,65] are possibly related with the elevated level of glutamine.

Urea has been determined as a cryoprotectant in the terrestrially hibernating wood frog Rana sylvatica [66]. Urea levels increase in the diapause-destined larvae of the cotton bollworm *Helicoverpa armigera* in the pre-diapause phase compared to their non-diapausing counterparts [67]. However, here we noted lower levels of urea in both the diapause and quiescence stages (Figure 6). Presumably, this could be due to the reduced metabolic activity of *S. mosellana* during these two stages and urea may not play a role in cold resistance in *S. mosellana*.

## 5. Conclusions

In summary, we identified significantly changed metabolites and metabolic pathways during the diapause process of *S. mosellana*. Reduced levels of TCA cycle intermediates (malic acid, citric acid, fumaric acid, α-ketoglutaric acid) and most amino acids in the larvae of both the diapause and quiescence stages may be mainly responsible for low metabolic activity, which enables this insect to survive under adverse temperature extremes in the hot summer and cold winter and also to synchronize its development with the wheat phenology. The significant elevation of trehalose, glycerol, mannitol, oleic acid, linoleic acid, proline, alanine and glutamine in the diapause or/and quiescence stages could serve as energy reserves for post-diapause development and/or protectants for cells and tissues during stresses and injuries. Overall, this study is the first to describe the changes in metabolite profiles during diapause in this key wheat pest and provides a more detailed insight into the metabolic adaptive mechanisms underlying diapause.

## Figures and Tables

**Figure 1 insects-13-00339-f001:**
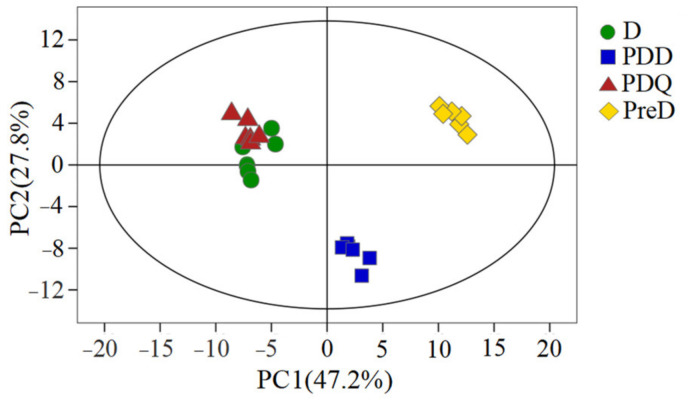
Principal component analysis (PCA) score plot of the metabolite profiles of *S. mosellana* larvae at pre-diapause (PreD), diapause (D), post-diapause quiescence (PDQ) and post-diapause development (PDD). Data points are displayed as sample projections onto the two primary axes. Oval represents the 95% confidence interval of the model using Hotelling T2 statistics. Variances explained by the first two components (PC1 and PC2) are shown in parentheses.

**Figure 2 insects-13-00339-f002:**
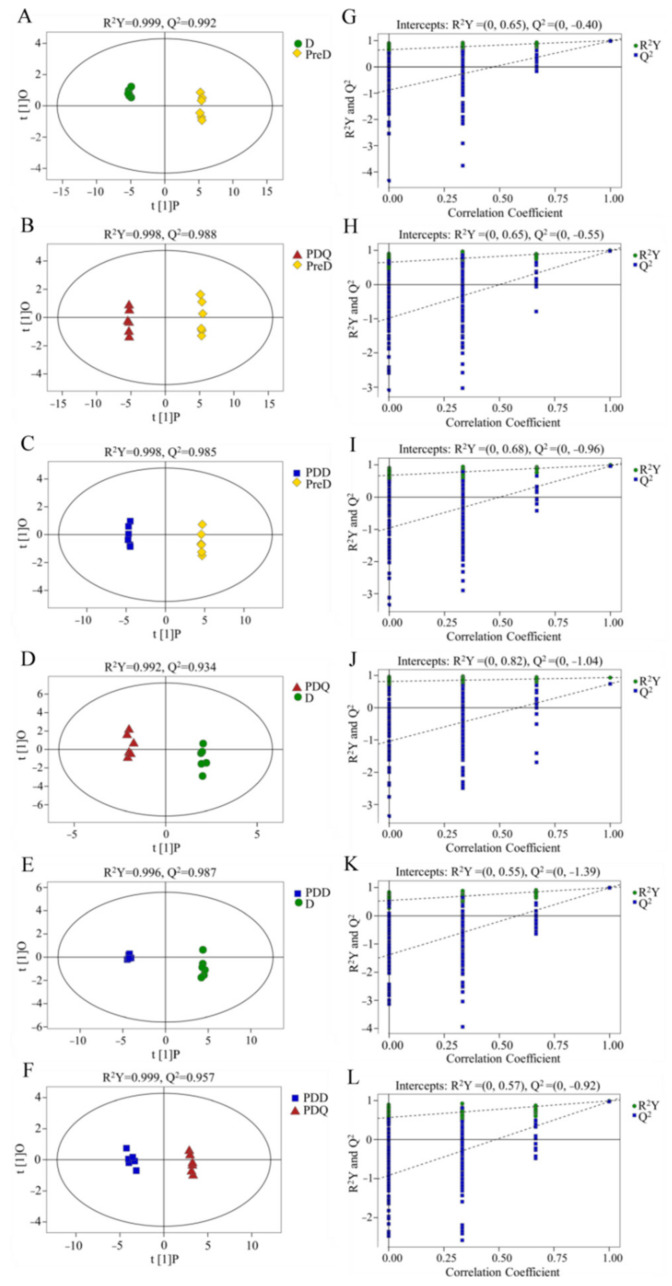
Orthogonal partial least squares discriminant analysis (OPLS-DA) score plots (**A**–**F**) and the corresponding permutation tests (*n* = 200) (**G**–**L**) of the metabolic profiles of pair comparisons of the four groups, including pre-diapause (PreD), diapause (D), post-diapause quiescence (PDQ) and post-diapause developmental larvae (PDD) of *Sitodiplosis mosellana*.

**Figure 3 insects-13-00339-f003:**
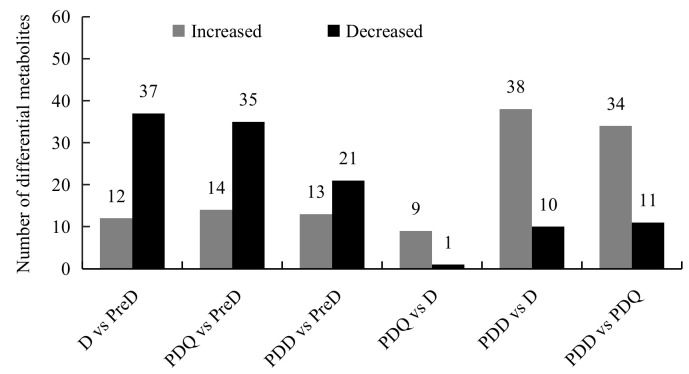
Number of differential metabolites of pairwise comparisons between pre-diapause (PreD), diapause (D), post-diapause quiescence (PDQ) and post-diapause developmental larvae (PDD) of *Sitodiplosis mosellana*.

**Figure 4 insects-13-00339-f004:**
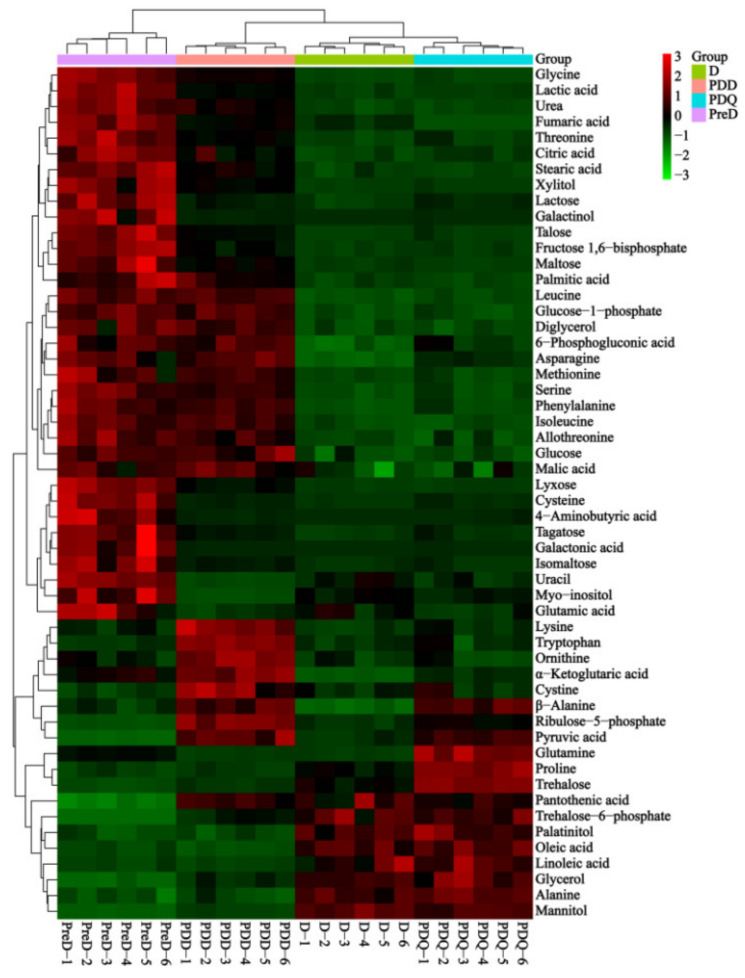
Hierarchical cluster analysis of all differential metabolites among pre-diapause (PreD), diapause (D), post-diapause quiescence (PDQ) and post-diapause developmental larvae (PDD) of *Sitodiplosis mosellana*. Red and green colors represent increased and decreased metabolites relative to the median metabolic level (black), respectively.

**Figure 5 insects-13-00339-f005:**
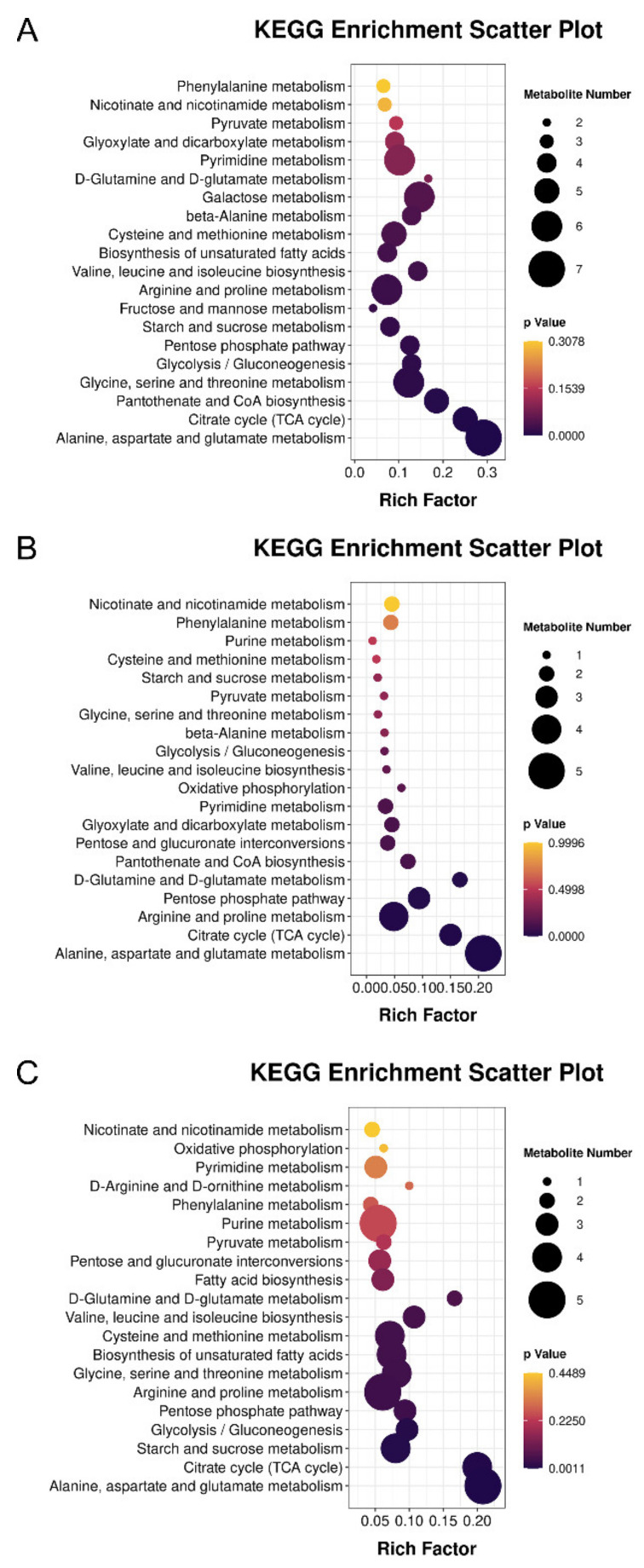
Pathway analysis of differential metabolites for three pairwise comparisons including “diapause versus pre-diapause” (**A**), “post-diapause quiescence versus diapause” (**B**) and “post-diapause development versus post-diapause quiescence” (**C**). The color and size of the shapes represent the effect of the diapause on the metabolism pathways. Large, dark shapes indicate a greater effect on the pathway.

**Figure 6 insects-13-00339-f006:**
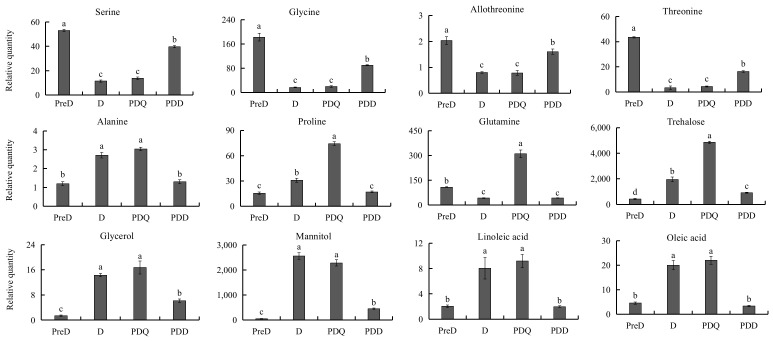
The normalized relative levels of representative metabolites in pre-diapause (PreD), diapause (D), post-diapause quiescent (PDQ) and post-diapause developmental larvae (PDD) of *Sitosiplosis mosellana*. Bars represent the means ± SE of six biological replicates and those with different letters were significantly different according to Tukey’s multiple range test (*p* < 0.05).

**Figure 7 insects-13-00339-f007:**
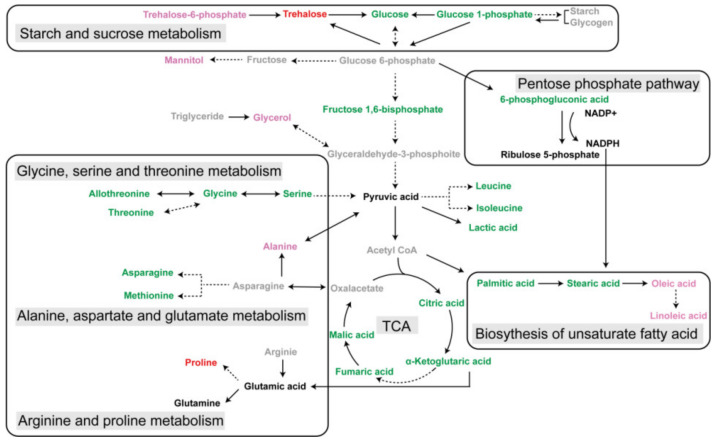
Relationships among the principal metabolic pathways of key differential metabolites during the diapause program of *Sitosiplosis mosellana*. Grey, no detection; green, lower at both diapause (D) and post-diapause quiescence stages (PDQ); pink and red, higher at both D and PDQ; red, the highest at PDQ.

## Data Availability

The data supporting reported results are available in the Appendix A of this article.

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
