# Peer review of "Metabolomics Reveals Changes in Metabolite Profiles among Pre-Diapause, Diapause and Post-Diapause Larvae of Sitodiplosis mosellana (Diptera: Cecidomyiidae)"

_insects, 2022, doi:10.3390/insects13040339_

Round 1
Reviewer 1 Report
|
Line |
Comment |
|
General comments |
- English language usage and grammar is well done - Structure throughout is mostly well done, with a couple tweaks needed - The paper would benefit most from some rewriting using an experimental framework with hypotheses and predictions relevant to the organism and its role as a pest, rather than the descriptive framework currently being used. - There’s some statistical wizardry making some pairwise comparisons seems more significant than they actually are. - It’s difficult to gage the novelty or usefulness of this paper as written. It follows a standard genomics/metabolics structure of describing all the things that changed, rather than those that are new or specifically interesting to this organism. The paper would benefit from focusing specifically on novel this type of result, rather than overall description and move some things to supplemental materials. |
|
50 |
A hallmark of diapause is metabolic suppression. This is a must, not something that sometimes accompanies diapause. |
|
61 |
Some reorganization of this paragraph needed- jumps around topics a bit. Is this supposed to be about diapause or the midge as a pest? Additionally, since this is a hot and cold diapausing insect, some background on any differences observed between the two states and their contribution to this organism’s role as a pest would be welcome and set you up for some interesting hypotheses and predictions. |
|
89 |
Are there any specific hypotheses or predictions based on the diapausing biology this organism? It would make the paper much stronger. |
|
101 |
How did you determine which were in diapause and which were in post-diapause quiescence? |
|
178 |
More explanation in the methods on how diapausing and post-diapause quiescence individuals were separated prior to metabolite analysis is needed. |
|
193 |
Just because a statistical method is capable of separating groups, does not mean those separations have biological significance. I am especially leery of the D/ PDQ separation with this method given the PCA did not yield those same results. Please explain whether you think this method is biologically relevant, and if not, it should probably be nixed.
|
|
253 |
This figure doesn't serve a specific purpose here. It would be better as part of supplemental materials. However, if you write the paper from a more experimental point of view, choosing specific panels from this figure for the main text would be much more useful. |
|
258 |
I'm not sure how useful this figure is either. It is obvious that the TCA cycle will be affected by diapause, so this doesn't add any novelty or interest to the findings of the paper. Kick it to supplemental. Again, reserve a pathway figure for only novel aspects or those that illustrate results relevant to specific hypotheses. |
|
271 |
unclear from the framing of the paper (intro) what the point of these 48 may be and whether they're specific to this organism or more generalizable/how they might be generalizable. |
|
274 |
These are useful for diapause in cold, but this midge also diapauses in hot dry weather. Some discussion of this aspect would be useful. |
|
285 |
recommend separating cold and warm diapause to different paragraphs |
|
286 |
What role do you think mannitol plays in warm diapause? |
|
344 |
Where's the discussion of your 48 new metabolites and their putative roles? |
Author Response
General Comments:
- English language usage and grammar is well done
Thanks.
- Structure throughout is mostly well done, with a couple tweaks needed.
Revisions (see below) have been made as suggested.
- The paper would benefit most from some rewriting using an experimental framework withhypotheses and predictions relevant to the organism and its role as a pest, rather than the descriptive framework currently being used.
We appreciate this reviewer’s constructive suggestions. We have made changes to paragraphs 2 and 4 in Introduction, also rewritten the paragraph 1 of Discussion.
- There’s some statistical wizardry making some pairwise comparisons seems more significant than they actually are.
PCA and OPLS-DA are often applied together to yield valuable insights on metabolomics datasets. Pairwise PCA is firstly performed to show the separation between two treatments, and then OPLS-DA can be conducted with high validity. Please see the publication “PCA as a practical indicator of OPLS-DA model reliability, Curr Metabolomics 2016, 4(2): 97-103. ”
In the previous manuscript, we did not show the figure of the pairwise PCA for the comparison between D and PDQ. Now, we added this figure as Figure S1. D and PDQ tended to group separately in the score plot of pairwise comparison although it was less obvious in the four-group plot. Therefore, both PCA and OPLS-DA are consistent, and our analysis likely reflected the biological difference.
- It’s difficult to gage the novelty or usefulness of this paper as written. It follows a standard genomics/metabolics structure of describing all the things that changed, rather than those that are new or specifically interesting to this organism. The paper would benefit from focusing specifically on novel this type of result, rather than overall description and move some things to supplemental materials.
Our study is unique because it has facilitated understanding of the biochemical adaptative mechanism underlying diapause and related stresses in this key wheat pest. Differing from many biochemical studies on insect diapause, insect samples at four developmental stages used here including pre-diapause, diapause, post-diapause quiescence and post-diapause development, were collected from field (rather than from the artificial lab setting), presumably more closely reflecting natural situation. This study identified several important metabolites/metabolic pathways during diapause of this midge, e.g. cryoprotectants alanine, proline and glutamine, as well as pentose phosphate pathway, biosynthesis pathway of unsaturated fatty acids.
As suggested, we modified the paragraphs 2 and 4 of Introduction, as well as the paragraph 1 of Discussion. So the structure of the manuscript was framed to the hypothesis-driven format.
Specific Comments:
- Line 50– A hallmark of diapause is metabolic suppression. This is a must, not something that sometimes accompanies diapause.
This sentence has been modified as suggested.
- Line 61– Some reorganization of this paragraph needed- jumps around topics a bit. Is this supposed to be about diapause or the midge as a pest? Additionally, since this is a hot and cold diapausing insect, some background on any differences observed between the two states and their contribution to this organism’s role as a pest would be welcome and set you up for some interesting hypotheses and predictions.
This paragraph has been rephrased to clarify differences between states of over-summering and over-wintering insects, and add the prediction for this study.
- Line 89– Are there any specific hypotheses or predictions based on the diapausing biology this organism? It would make the paper much stronger.
This paragraph has been reorganized as suggested.
- Line 101– How did you determine which were in diapause and which were in post-diapause quiescence?
This midge undergoes obligatory larval diapause in soil inside the cocoon. Our previous work showed that almost all cocooned larvae collected in December or later could arouse and emerge as adults once exposed to favorable temperature (25℃), indicating that by December they have completed diapause and entered post-diapause quiescence. In contrast, no larvae collected in summer (June to August) and very few larvae collected in autumn (September to November) were able to emerge, suggesting they are in the phase of diapause (Cheng et al., 2017).
Information has now been included in the text.
- Line 178- More explanation in the methods on how diapausing and post-diapause quiescence individuals were separated prior to metabolite analysis is needed.
Explanation has been added in the method (Section 2.1).
- Line 193– Just because a statistical method is capable of separating groups, does not mean those separations have biological significance. I am especially leery of the D/ PDQ separation with this method given the PCA did not yield those same results. Please explain whether you think this method is biologically relevant, and if not, it should probably be nixed.
We believe that the statistical difference is biologically relevant. See comment so response to general comment #4.
- Line 253– This figure doesn't serve a specific purpose here. It would be better as part of supplemental materials. However, if you write the paper from a more experimental point of view, choosing specific panels from this figure for the main text would be much more useful.
Thanks very much for your advice. We choose several key metabolites potentially important for diapause of this insect from this figure for the main text.
- Line 258– I'm not sure how useful this figure is either. It is obvious that the TCA cycle will be affected by diapause, so this doesn't add any novelty or interest to the findings of the paper. Kick it to supplemental. Again, reserve a pathway figure for only novel aspects or those that illustrate results relevant to specific hypotheses.
We have modified this figure, and highlighted the important findings including novel biological characteristics for S. mosellana diapause (e.g. the pentose phosphate pathway, biosynthesis pathway of unsaturated fatty acids). This figure summarized the key findings in this study and we feel it is useful for readers to understand the take-home message.
- Line 271 – unclear from the framing of the paper (intro) what the point of these 48 may be and whether they're specific to this organism or more generalizable/how they might be generalizable.
This paragraph has been revised. Please see paragraph 1 in Discussion.
- Line 274– These are useful for diapause in cold, but this midge also diapauses in hot dry weather. Some discussion of this aspect would be useful.
We discussed the role of glycerol and trehalose in diapause and cold acclimation of this midge based on their high levels in diapause and particularly in cold quiescence stage: (1) serving as important energy reserves; (2) acting as cryoprotectants by stabilizing proteins and membranes and reducing the body’s supercooling point. Presumably, they do not play important roles in heat/drought tolerance.
As a typical insect undergoing obligatory diapause, diapause initiation is programmed, unrelated to environmental factors (e.g. temperature, photoperiod), differing from facultative diapause. Previous studies have shown that synthesis of these compounds can be promoted by diapause itself, and thus their high levels in over-summering diapausing larvae of S. mosellana may not be related with hot/dry weather unless they are higher than in cold winter. It's worth noting that chilling is necessary for diapause termination and quiescence of this midge, and thus high levels of these two compounds may be involved in cold tolerance.
- Line 285– recommend separating cold and warm diapause to different paragraphs
Done as suggested.
- Line 286– What role do you think mannitol plays in warm diapause?
Mannitol may serve as energy reserves and thermo-osmoprotectant in over-summering diapausing individuals by stabilizing proteins and preserving cell membrane fluidity.
Information has been incorporated into the text.
- Line 344– Where's the discussion of your 48 new metabolites and their putative roles?
We identified 48 new metabolites. However, not all metabolites have clear functions for insects and some were low in the diapause stage. So the discussion mainly focused on the metabolites with known functions in other organisms. We have reorganized the paragraph 1 of Discussion.

Reviewer 2 Report
The manuscript is well written and easy to follow. However, there are so many abbreviations throughout of the manuscript, which makes it time-consuming to read through by back and forth checking abbreviations. Usually authors avoid to use abbreviation unless the terms are well known such as polymerase chain reaction (PCR). Another suggestion is to avoid up-regulated or down-regulated, which are two terms used for gene expression levels. Regulated means action. The changes in metabolite levels may simply due to passive events, not actively regulated.
Author Response
The manuscript is well written and easy to follow. However, there are so many abbreviations throughout of the manuscript, which makes it time-consuming to read through by back and forth checking abbreviations. Usually authors avoid to use abbreviation unless the terms are well known such as polymerase chain reaction (PCR). Another suggestion is to avoid up-regulated or down-regulated, which are two terms used for gene expression levels. Regulated means action. The changes in metabolite levels may simply due to passive events, not actively regulated
In studies on insect diapause, physiological stages are usually abbreviated in the text (Khodayari et al., 2013; Wang et al., 2017 in the manuscript). In the revised version we used the full term of four developmental stages except for those in Results because of high frequency of occurrence in this section. We also replaced DM with “different metabolite”.
We replaced “up-regulated” or “down-regulated” in the text as suggested.
Round 2
Reviewer 1 Report
Thank you for addressing all comments adequately. Much improved!